# A Pilot Study of the Adverse Events Caused by the Combined Use of Bevacizumab and Vascular Endothelial Growth Factor Receptor-Targeted Vaccination for Patients with a Malignant Glioma

**DOI:** 10.3390/vaccines8030498

**Published:** 2020-09-02

**Authors:** Ryota Tamura, Yukina Morimoto, Mizuto Sato, Tetsuro Hikichi, Kazunari Yoshida, Masahiro Toda

**Affiliations:** 1Department of Neurosurgery, Keio University School of Medicine, Tokyo 160-8582, Japan; moltobello-r-610@hotmail.co.jp (R.T.); yukinaxnashiko@yahoo.co.jp (Y.M.); mizuto.sato@gmail.com (M.S.); kazrmky@keio.jp (K.Y.); 2OncoTherapy Science Inc., 3-2-1, Sakado, Takatsu-ku, Kawasaki City, Kanagawa 213-0012, Japan; t-hikichi@cancerprecision.co.jp

**Keywords:** bevacizumab, VEGF-A, VEGFR, peptide vaccine, adverse event, malignant glioma

## Abstract

Anti-angiogenic therapy, targeting vascular endothelial growth factor (VEGF)-A/VEGF receptors (VEGFRs), is beneficial for tumor growth prevention in a malignant glioma. A simultaneous blockade using both bevacizumab (Bev), which targets circulating VEGF-A, and a multi-kinase inhibitor on VEGFRs was more effective for advanced solid cancers, including melanoma and renal cell carcinoma. However, previous clinical trials demonstrated a high adverse event rate. Additionally, no studies previously assessed treatment efficacy and safety using both VEGF-A and VEGFR-targeted agents for malignant gliomas. We had conducted clinical trials investigating VEGFRs peptide vaccination in patients with malignant gliomas, in which the treatment exhibited safety and yielded therapeutic effects in some patients. The combined use of Bev and VEGFRs vaccination may enhance the anti-tumor effect in malignant gliomas. In this pilot study, the adverse event profile in patients treated with Bev after the vaccination was investigated to establish this treatment strategy, in comparison to those treated with Bev collected from the published data or treated with the vaccination alone. In our previous clinical studies on patients with malignant gliomas, Bev was administered to 13 patients after VEGFRs vaccinations. One patient had a Grade 4 pulmonary embolism. Two patients had Grade 2 cerebral infarctions. There were no significant differences in the adverse event rates among patients treated with Bev, with the vaccination, or with Bev after the vaccination. Although careful observation is imperative for patients after this combination treatment strategy, VEGFRs-targeted vaccination may coexist with Bev for malignant gliomas.

## 1. Introduction

A glioblastoma is the most aggressive primary brain tumor. The current standard treatment after surgical resection, which includes radiotherapy and chemotherapy with temozolomide, provides a limited degree of improvement in survival rate, with a median overall survival of 15 months [1]. Glioblastomas exhibit extensive vascularity. The vascular endothelial growth factor (VEGF)-A and VEGF receptor (VEGFR) signaling are strongly upregulated, and the degree of expression correlates with the grade of malignancy and prognosis of the glioblastomas [2,3,4,5]. Therefore, VEGF-A/VEGFR signaling seems to be an attractive target for anti-angiogenic therapy in glioblastomas. Bevacizumab (Bev), which targets circulating VEGF-A; multi-kinase inhibitors such as cediranib, which target VEGFRs, platelet-derived growth factor receptors (PDGFRs), and c-kit; sunitinib, which targets VEGFRs and PDGFRs; and sorafenib, which targets VEGFRs, PDGFRs, Raf, and c-kit, have all been used to treat glioblastomas. They were associated with favorable event-free survival and improvement of peritumoral edema, leading to a better patient performance status [6,7,8,9,10,11,12,13]. Bev is currently used most frequently in the management of glioblastomas [14,15]. However, clinical trials using these anti-angiogenic agents have suggested that the favorable clinical effects on patients’ performance status do not translate into an overall survival benefit. VEGF/VEGFRs-targeted therapy is known to enhance the effects of immunotherapy. VEGF-A/VEGFR signaling suppresses the antitumor immune response by inhibiting the maturation of dendritic cells and stimulating the proliferation of regulatory T cells [16,17,18,19]. A recent randomized trial demonstrated the anti-programmed death 1 (PD-1) antibody nivolumab’s failure to prolong overall survival in patients with recurrent glioblastomas [20]. The use of a combination of an anti-angiogenic agent and an immune checkpoint inhibitor may be a strategy for overcoming the mechanism of resistance of immunotherapies for glioblastomas [21]. Therefore, anti-angiogenic therapy will remain a critical treatment option for patients with glioblastomas.

We have previously conducted exploratory clinical trials investigating VEGFRs peptide vaccination with and without multiple glioma oncoantigens, such as lymphocyte antigen 6 family member K (LY6K), DEP domain containing 1 (DEPDC1), kinesin family member 20A (KIF20A), and forkhead box M1 (FOXM1), in patients with primary and recurrent malignant gliomas, wherein treatment exhibited safety and yielded therapeutic effects in some patients [14,22,23]. Recently, in our hospital, a clinical trial using a VEGFRs peptide vaccine was also conducted in patients with progressive neurofibromatosis type 2 (NF2)-derived schwannomas, showing hearing improvement and tumor volume reduction [15].

Memory-induced cytotoxic T lymphocytes (CTLs) after the VEGFRs vaccination may persistently suppress tumor progression, and dual VEGF/VEGFR inhibition has the possibility to enhance the anti-tumor effect and anti-tumor immune response [16,24]. The simultaneous blockade of both VEGF-A and VEGFRs, using Bev and multi-kinase inhibitors, was reported to be more clinically efficacious for patients with advanced solid cancers, including melanoma, and renal cell carcinoma [25,26,27,28,29]. However, in those studies, a high degree of hypertension, thrombocytopenia, and thrombotic microangiopathy was observed. Therefore, these adverse events precluded the further development of this combinational strategy with Bev [28,29]. We must be careful in adding Bev to other treatment strategies, because the combined approach using Bev and interferon alfa-2a demonstrated that a higher advanced event rate was observed in Bev-treated patients with renal cell carcinoma compared with those without Bev [30].

To establish the treatment strategy with the combined use of Bev and VEGFRs peptide vaccination for malignant gliomas, we need to evaluate the adverse event profile of Bev on patients after the vaccination.

## 2. Materials and Methods

We have previously conducted exploratory clinical trials investigating VEGFRs peptide vaccination with and without multiple glioma oncoantigens (LY6K, DEPDC1, KIF20A, and FOXM1) in patients with primary and recurrent malignant gliomas (UMIN000013381 (jRCTs031180170), UMIN000012774 (jRCTs031180148), and UMIN000005545) [14,22,23]. The inclusion and exclusion criteria were shown in our previous studies [14,22,23]. The Keio University School of Medicine Ethics Committee approved the trials, which were conducted in accordance with the Helsinki declaration on experimentation on human subjects. The authors affirm that human research participants provided informed consent to participate in the study and the publication of their data (see each reference).

For this retrospective pilot case-control study, the patients treated with Bev after the VEGFRs peptide vaccination (Vac + Bev group; *n* = 13, 17–78 y/o, 8 males and 5 females) were selected from our three previous clinical trials (May 2012 to March 2018). In addition, we collected data from patients who received only a VEGFR vaccination (without Bev) (Vac group; *n* = 15, 34–75 y/o, 7 males and 8 females) (May 2012 to March 2018). Bevacizumab was administered at 10 mg/kg every 2 weeks. Some clinical information (type of vaccination, timing of Bev administration after the vaccination, number of vaccinations) and complications were evaluated. No studies assessing the treatment effect for malignant gliomas using both VEGF-A and VEGFR-targeted agents have been reported previously. Therefore, the adverse event profiles of the Vac and Vac + Bev groups were compared with the previously reported adverse event profile of the 55 patients with malignant glioma treated by Bev alone (Bev group) [31]. An adverse event was assessed using the Common Terminology Criteria for Adverse Events version 4.0. None of the patients had a potentially relevant past medical history, such as vasculopathy, in the Vac and Vac + Bev groups. Data from CTL induction were also collected, which were evaluated by the contract research organizations “OncoTherapy Science, Inc. (Kanagawa, Japan) and “Cancer Precision Medicine, Inc. (Kanagawa, Japan)””, as previously described [15,32]. To assess the relationship between the clinical information and the adverse events in the Vac + Bev group, and to evaluate the adverse event rate among these groups, the chi-squared test was used. Student’s t-test and the chi-squared test were used to evaluate the patients’ general characteristics between Vac and Vac + Bev groups. All statistical analyses were performed with IBM SPSS software (IBM Corp., Armonk, NY, USA). A *p*-value of <0.05 was considered to be statistically significant.

## 3. Results

No significant differences were identified in the patients’ general characteristics between the Vac and Vac + Bev groups (Table 1). In the Vac + Bev group, Bev was administered to 13 patients after the VEGFRs vaccinations. The median number of VEGFRs vaccine and Bev administrations was 10 and 6, respectively. The median timing of initiation of Bev administration after the last vaccination was 13 days. One patient with recurrent glioblastoma had a Grade 4 pulmonary embolism. Two patients with recurrent glioblastomas had Grade 2 cerebral infarctions during Bev administration. There were no other adverse events (>Grade 2) (Table 2). In the Vac + Bev group, the chi-squared test was used to compare categorical variables (type of vaccination, VEGFRs vs. VEGFRs with multiple glioma oncoantigens; timing of Bev administration, ≥110 days vs. <110 days after the last vaccination; number of vaccinations, ≥10 vs. <10; CTL induction of VEGFR1 or VEGFR2, positive vs. negative). In the present study, the mean value was used to determine the cut-off value, as previously described [33]. However, there was no statistically significant association between type of vaccination, number of vaccinations, CTL induction, timing of Bev administration, and complications (*p* = 0.91, *p* = 0.24, *p* = 0.13, *p* = 0.76, respectively).

There were no Grade 3 adverse events in the Vac group. Grade 2 cerebral infarction was observed in two patients with recurrent glioblastomas during vaccination (Table 3). In the Bev group [31], the number of adverse events was 1 (Grade 2 hypertension), 1 (Grade 2 epistaxis), 1 (Grade 2 phlebitis), 1 (Grade 2 impaired wound healing), 1 (Grade 3 deep venous thrombosis), 1 (Grade 4 colon perforation), and 4 (Grade 4 pulmonary embolism) [31]. No significant differences were identified in the adverse event rate among the three groups (≥Grade 2, Vac vs. Vac + Bev, *p* = 0.50; Bev vs. Vac + Bev, *p* = 0.69; Vac vs. Bev, *p* = 0.66: ≥Grade 3, Vac vs. Vac + Bev, *p* = 0.27; Bev vs. Vac + Bev, *p* = 0.73; Vac vs. Bev, *p* = 0.18) (Figure 1).

## 4. Discussion

Although the simultaneous blockade of both VEGF-A and VEGFRs exhibited a further anti-tumor effect, previous phase I trials with combined usage of sunitinib and Bev demonstrated higher adverse event rates, including arterial adverse events, particularly cardiac and cerebral ischemia, venous adverse events, bleeding, and arterial hypertension for advanced solid cancers [6,8,9,25,34]. In the present analysis, three out of 13 patients with a malignant glioma who used a VEGFRs peptide vaccine exhibited ≥ Grade 2 complications after Bev administration. There were no significant differences in the incidence of adverse events among the Vac, Bev, and Vac + Bev groups. Therefore, the adverse event profile of Bev in patients after the vaccination may be similar to VEGFR vaccination or Bev monotherapy. In addition, no unexpected adverse events related to Bev monotherapy were observed in the combination treatment. Although no studies have previously evaluated the efficacy and safety of combinational therapy using both VEGF-A and VEGFR-targeted agents for malignant gliomas, VEGFRs vaccination with Bev has the possibility of being applied in the treatment of malignant gliomas. However, patients on the combination treatment strategy should be under close monitoring of blood pressure, blood chemistry, and urine protein to minimize the risk of serious adverse events.

Furthermore, this combination treatment strategy may be useful for other refractory hypervascular tumors. Bev has proved to be efficacious for the treatment of patients with neurofibromatosis 2 (NF2) [35,36,37]. Our exploratory clinical trial also demonstrated that VEGFRs peptide vaccination showed hearing improvement and tumor volume reduction in NF2 patients. There were no severe adverse events related to the vaccine [15]. Recently, we initiated another clinical trial using a VEGFRs peptide vaccine for recurrent, progressive, and refractory benign brain tumors (hemangiopericytoma, hemangioblastoma, meningioma, chordoma, and ependymoma). The adverse event profile of Bev after VEGFRs vaccination must be evaluated for these tumors as well as malignant gliomas.

A limitation of the present pilot study was the paucity of the number of patients treated by Bev after the VEGFRs peptide vaccination. A small sample size decreases the statistical power. Malignant gliomas treated by Bev administration after VEGFRs vaccination are rare. Because detailed data on the Bev group could not be obtained from the previous report, statistical analysis was not performed to evaluate the patients’ general background among Vac, Bev, and Vac + Bev groups. In addition, it remains unclear whether this combination treatment strategy can improve the overall survival in patients with malignant gliomas. Studies using a larger number of patients with various types of tumors are warranted to confirm and generalize the findings of this study.

## 5. Conclusions

VEGFRs-targeted vaccination with Bev may be an option that leads to a novel approach to the treatment of patients with malignant gliomas. A further randomized prospective study is needed to confirm the present findings.

## Figures and Tables

**Figure 1 vaccines-08-00498-f001:**
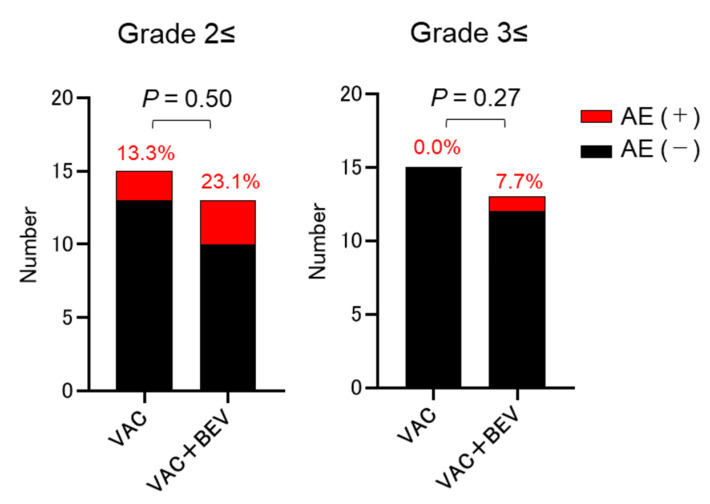
Comparison of the percentage of adverse events in malignant glioma patients treated with vaccination and with both vaccination and bevacizumab. AE, adverse event; Bev, bevacizumab; Vac, vaccination.

**Table 1 vaccines-08-00498-t001:** Summary of patients’ general characteristics in Vac and Vac + Bev groups.

	Vac Group	Vac + Bev Group	*p* Value
Age (mean ± SD (med))	55.07 ± 12.35 (59)	49.69 ± 16.67 (47)	0.17
Sex	M: 7, F: 8	M: 8, F: 5	0.43
GB	Primary: 4, Recurrent: 11	Primary: 2, Recurrent: 11	0.47
Type of Vaccination	VEGFR1/2: 9, Cocktail ^※1^: 6	VEGFR1/2: 4, Cocktail ^※1^: 9	0.12
Number of Vaccinations (mean ± SD (med))	12.47 ± 3.70 (12)	10.69 ± 5.14 (10)	0.16

F, female; GB, glioblastoma; M, male; med, median; VEGFR, Vascular Endothelial Growth Factor Receptor. Cocktail ^※1^: VEGFRs peptide vaccination with multiple glioma oncoantigens (LY6K, DEPDC1, KIF20A, and FOXM1.

**Table 2 vaccines-08-00498-t002:** Adverse event profile of patients treated with Bev after VEGFRs peptide vaccination (Vac + Bev group).

Case	Age/Sex	PH	Diagnosis (IDH/MGMT Methylation)	Type of Vac	CTL Induction (VEGFR1/VEGFR2)	Number of Vac Administrations	Timing of Bev after Vac (days)	Number of Bev Administrations	Complication
1	76/M	Lacunar infarction	Primary GB (WT/unmethyl)	VEGFR1/2	NT	16	461	6	-
2	50/F	-	Primary GB (WT/unmethyl)	VEGFR1/2	NT	2	76	5	-
3	41/F	-	Recurrent high-grade glioma	VEGFR1/2	+/+	8	13	4	-
4	37/M	-	Recurrent high-grade glioma	VEGFR1/2	−/−	17	598	11	PE (Grade 4)
5	17/M	-	Recurrent high-grade glioma (WT/unmethyl)	Cocktail ^※1^	+/−	18	1	2	-
6	38/M	-	Recurrent high-grade glioma	Cocktail	−/−	11	7	9	CI (Grade 2)
7	47/F	-	Recurrent GB	Cocktail	+/+	8	7	3	-
8	73/M	-	Recurrent high-grade glioma (WT/unmethyl)	Cocktail	NT	8	42	1	-
9	68/F	Asthma Hyperlipidemia	Recurrent GB (WT/unmethyl)	Cocktail	NT	18	7	14	-
10	61/M	Hyperlipidemia	Recurrent GB (WT/methyl)	Cocktail	−/+	6	1	6	-
11	38/F	-	Recurrent GB (WT/methyl)	Cocktail	NT	11	35	9	CI (Grade2)
12	55/M	Nasal sinuses inflammation	Recurrent GB (WT/unmethyl)	Cocktail	NT	6	168	3	-
13	45/M	-	Recurrent GB (WT/unmethyl)	Cocktail	NT	10	8	9	-

Bev, bevacizumab; CI, cerebral infarction; CTL, cytotoxic T lymphocyte; F, female; GB, glioblastoma; IDH, isocitrate dehydrogenase; M, male; methyl, methylation; MGMT, O6- methylguanine-DNA methyltransferase; NT, not tested; PE, pulmonary embolism; PH, past history; unmethyl, unmethylation; Vac, vaccination; VEGFR, Vascular Endothelial Growth Factor Receptor; WT, wild-type. ^※1^ Cocktail: VEGFRs peptide vaccination with multiple glioma oncoantigens (LY6K, DEPDC1, KIF20A, and FOXM1).

**Table 3 vaccines-08-00498-t003:** Adverse event profile of patients after VEGFRs peptide vaccination without Bev (Vac group).

Case	Age/Sex	PH	Diagnosis (IDH/MGMT Methylation)	Type of Vac	CTL Induction (VEGFR1/VEGFR2)	Number of Vac Administrations	Complication
1	52/M	-	Primary GB (WT/unmethyl)	VEGFR1/2	+/+	14	-
2	64/M	-	Primary GB (WT/unmethyl)	VEGFR1/2	NT/NT	5	-
3	50/F	Pituitary adenoma Asthma	Primary GB (WT/methyl)	VEGFR1/2	+/+	14	-
4	60/F	-	Primary high-grade glioma (WT/methyl)	VEGFR1/2	NT/NT	14	-
5	75/F	-	Recurrent GB (WT/NA)	VEGFR1/2	+/−	10	CI (Grade 2)
6	62/F	Duodenal ulcer	Recurrent high-grade glioma (WT/unmethyl)	VEGFR1/2	+/−	11	-
7	68/M	-	Recurrent high-grade glioma (mutant/mechyl)	VEGFR1/2	+/−	12	-
8	59/M	Ischemic heart disease Colon polyp	Recurrent GB (WT/NA)	VEGFR1/2	+/−	8	-
9	62/M		Recurrent GB (WT/unmethyl)	VEGFR1/2	+/−	10	-
10	38/M	-	Recurrent GB (WT/methyl)	Cocktail ^※1^	+/−	18	-
11	66/F	-	Recurrent GB (NA/NA)	Cocktail	−/−	12	-
12	34/F	-	Recurrent high-grade glioma (mutant/unmethyl)	Cocktail	+/+	20	-
13	36/F	-	Recurrent GB (mutant/unmethyl)	Cocktail	NT/NT	11	-
14	55/M	-	Recurrent GB (WT/unmethyl)	Cocktail	NT/NT	14	CI (Grade 2)
15	45/F	-	Recurrent high-grade glioma (mutant/methyl)	Cocktail	NT/NT	14	-

Bev, bevacizumab; CI, cerebral infarction; CTL, cytotoxic T lymphocyte; F, female; GB, glioblastoma; IDH, isocitrate dehydrogenase; M, male; methyl, methylation; MGMT, O6- methylguanine-DNA methyltransferase; NA, not available; NT, not tested; PH, past history; unmethyl, unmethylation; Vac, vaccination; VEGFR, Vascular Endothelial Growth Factor Receptor; WT, wild-type. Cocktail ^※1^: VEGFRs peptide vaccination with multiple glioma oncoantigens (LY6K, DEPDC1, KIF20A, and FOXM1).

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
