# Peer review of "A Pilot Study of the Adverse Events Caused by the Combined Use of Bevacizumab and Vascular Endothelial Growth Factor Receptor-Targeted Vaccination for Patients with a Malignant Glioma"

_vaccines, 2020, doi:10.3390/vaccines8030498_

Round 1
Reviewer 1 Report
Thank you for the revised manuscript. This paper has been significantly improved, however, some concerns still exist.
- What was the dosing regimen for Bev?
- The comparability of current patients treated with Bev and VEGFRs vaccinations and a previous cohort treated only with Bev was not commented on by the authors. This is an important area of concern.
- What is meant by "careful observation"? What should we observe for?
Author Response
Point by point response
We are very grateful to the reviewers for their insightful comments and suggestions, which have undoubtedly helped us to improve our manuscript immensely. As indicated in the responses below, we have taken all their comments and suggestions into account when generating the revised version of the manuscript. Responses to the reviewers’ comments appear after the arrows in blue text.
Reviewer 1:
Thank you for the revised manuscript. This paper has been significantly improved, however, some concerns still exist.
→
Thank you very much for your comments.
1.What was the dosing regimen for Bev?
→
Thank you for your comments.
Bevacizumab was administered at 10 mg/kg every 2 weeks. We have added this sentence in the revised manuscript.
- The comparability of current patients treated with Bev and VEGFRs vaccinations and a previous cohort treated only with Bev was not commented on by the authors. This is an important area of concern.
→
As the reviewer indicated, we have added further results and discussion.
No significant differences were identified in the adverse event rate among the three groups (≥ Grade 2, Vac vs. Vac+Bev, P=0.50; Bev vs. Vac+Bev, P=0.69; Vac vs. Bev, P=0.66: ≥Grade 3, Vac vs. Vac+Bev, P=0.27; Bev vs. Vac+Bev, P=0.73; Vac vs. Bev, P=0.18) (Figure 1).
There were no significant differences in the incidence of adverse events among the Vac, Bev and Vac+Bev groups. Therefore, the adverse event profile of Bev in patients after the vaccination may be similar to VEGFR vaccination or Bev monotherapy. No unexpected adverse events related to Bev monotherapy were observed by the combination treatment.
- What is meant by "careful observation"? What should we observe for?
→
According to the reviewer’s comment, we have added the following sentences.
Patients on the combination treatment strategy should be under close monitoring of blood pressure, blood chemistry, and urine protein to minimize the risk of serious adverse events.
Reviewer 2 Report
The authors have addressed a majority of my concerns, however, there are still some minor issues that needs some work.
- Introduction: Please provide some examples of multi-kinase inhibitors that are being used to treat malignant gliomas in the clinic.
- Lines 68-70 – Is there a reference for this statement?
- Line 95: When the authors say “frequency of vaccination”, what does they mean exactly? Is it in number of days or weeks? Please provide a proper clarification.
- Lines 111-112: “The median frequency of VEGFRs vaccine and Bev were 10 and 6, respectively.” What does this mean? Please rephrase this sentence.
- Issue #14 from 1st review & line 118-119: The authors state that they have used chi-squared test to determine associations between various parameters, but do find any statistically significant associations. It is just not enough to state that there were no associations. Can the authors provide the numerical data from these tests that prove their statement?
Author Response
Point by point response
We are very grateful to the reviewers for their insightful comments and suggestions, which have undoubtedly helped us to improve our manuscript immensely. As indicated in the responses below, we have taken all their comments and suggestions into account when generating the revised version of the manuscript. Responses to the reviewers’ comments appear after the arrows in blue text.
Reviewer 2:
The authors have addressed a majority of my concerns, however, there are still some minor issues that needs some work.
→
Thank you very much for your comments.
- Introduction: Please provide some examples of multi-kinase inhibitors that are being used to treat malignant gliomas in the clinic.
→
VEGF-A/VEGFRs signaling seems to be an attractive target of anti-angiogenic therapy in glioblastomas. Bevacizumab (Bev), which targets circulating VEGF-A, and multi-kinase inhibitor, such as cediranib, which targets VEGFR, platelet-derived growth factor receptor (PDGFR), and c-kit; sunitinib, which targets VEGFR and PDGFR; and sorafenib, which targets VEGFR, PDGFR, Raf, and c-kit, have been used to treat glioblastomas. They were associated with favorable event-free survival and improvement of peritumoral edema, leading to better patient performance status [6-13]. Bev is currently used more frequently in the management of glioblastomas [12,13]. However, clinical trials using these anti-angiogenic agents have suggested that the favorable clinical effects on patient’s performance status do not translate into an overall survival benefit.
- Lines 68-70 – Is there a reference for this statement?
→
Thank you for your comments. This is the theoretical hypotheses based on the previous knowledge. Theoretically, memory induced cytotoxic T lymphocytes (CTLs) after the VEGFRs vaccination may persistently suppress tumor progression, and dual VEGF/VEGFR inhibition has the possibility to enhance the anti-tumor effect and anti-tumor immune response [Disis ML. Semin Oncol. 2014; Tamura R. Med Oncol. 2020].
Reference
Disis, M.L.; Mechanism of action of immunotherapy. Semin Oncol. 2014, 5, S3-13.
Tamura, R.; Tanaka, T.; Akasaki, Y.; Murayama, Y.; Yoshida, K.; Sasaki, H: The role of vascular endothelial growth factor in the hypoxic and immunosuppressive tumor microenvironment: perspectives for therapeutic implications. Med Oncol. 2020, 37,2.
- Line 95: When the authors say “frequency of vaccination”, what does they mean exactly? Is it in number of days or weeks? Please provide a proper clarification.
→
Frequency of vaccination means the “number of vaccination”. For example, peptides were injected subcutaneously at infra-axillary and inguinal lymph nodes four times every week, and then four times monthly thereafter (eight times in total).
As previously described, we have used the “number of vaccination” in the revised manuscript [Iinuma H. J Transl Med. 2014; Hazama S. J Transl Med. 2014].
Reference
Iinuma, H., Fukushima, R., Inaba, T., Tamura, J., Inoue, T., Ogawa, E., Horikawa, M., Ikeda, Y., Matsutani, N., Takeda, K., et al; Phase I clinical study of multiple epitope peptide vaccine combined with chemoradiation therapy in esophageal cancer patients. J Transl Med. 2014,12, 84.
Hazama, S., Nakamura, Y., Takenouchi, H., Suzuki, N., Tsunedomi, R., Inoue, Y., Tokuhisa, Y., Iizuka, N., Yoshino, S., Takeda, K., et al; A phase I study of combination vaccine treatment of five therapeutic epitope-peptides for metastatic colorectal cancer; safety, immunological response, and clinical outcome. J Transl Med. 2014,12,63.
- Lines 111-112: “The median frequency of VEGFRs vaccine and Bev were 10 and 6, respectively.” What does this mean? Please rephrase this sentence.
→
According to the reviewer’s comments, we have modified the indicated sentence in the revised manuscript.
The median number of VEGFRs vaccination and Bev administration were 10 and 6, respectively.
- Issue #14 from 1st review & line 118-119: The authors state that they have used chi-squared test to determine associations between various parameters, but do find any statistically significant associations. It is just not enough to state that there were no associations. Can the authors provide the numerical data from these tests that prove their statement?
→
We have added the following sentences in the section of Results.
The Chi-squared test was used to compare categorical variables (type of vaccination, VEGFRs vs. VEGFRs with multiple glioma oncoantigen; timing of Bev administration, ≥110 days vs. <110 days after the last vaccination; number of vaccination, ≥10 vs. 10<; CTL induction of VEGFR1 or VEGFR2, positive vs. negative). In the present study, the mean value was used to determine the cut-off value. However, there was no statistically significant association between type of vaccination, number of vaccination, CTL induction, timing of Bev administration, and complications (P=0.91, P=0.24, P=0.13, P=0.76, respectively).
Reviewer 3 Report
In this new version of their MS Tamura et al have modified the statistically analysis to better compare their 2 cohorts of patients. Although, I recognize that the new version is improved, I consider that the results are not enough consistent to warrant publication in Vaccines. I am not an expert in statistical analysis, hence it is possible that I am wrong, but the low number of patients and the differences inherent to small cohorts preclude in my opinion a clear conclusion for this analysis.
Author Response
Point by point response
We are very grateful to the reviewers for their insightful comments and suggestions, which have undoubtedly helped us to improve our manuscript immensely. As indicated in the responses below, we have taken all their comments and suggestions into account when generating the revised version of the manuscript. Responses to the reviewers’ comments appear after the arrows in blue text.
Reviewer 3:
In this new version of their MS Tamura et al have modified the statistically analysis to better compare their 2 cohorts of patients. Although, I recognize that the new version is improved, I consider that the results are not enough consistent to warrant publication in Vaccines. I am not an expert in statistical analysis, hence it is possible that I am wrong, but the low number of patients and the differences inherent to small cohorts preclude in my opinion a clear conclusion for this analysis.
→
As we have noted in the section of limitation, the main limitation of the present study was the paucity of the number of patients treated by Bev after the VEGFRs peptide vaccination. However, malignant gliomas treated by Bev administration after VEGFRs vaccination are extremely rare. Small sample size decreases statistical power. Therefore, we have tempered the conclusions. We regret that it is very difficult to increase the number of cases. We believe that the conclusions of the manuscript will not change by the addition of a few more cases. We would be highly grateful if the revised manuscript is considered for publication in Vaccines.
Reviewer 4 Report
In the manuscript entitled “Adverse events caused by the combined use of bevacizumab and vascular endothelial growth factor receptor-targeted vaccination for the patients with malignant glioma”, the authors investigate adverse event profile in patients treated with Bev after the vaccination in comparison to those treated with Bev collected from the published data, or the vaccination alone. They did not find any significant differences in the adverse event rates between the three groups. Given the clinical relevance of the topic, the proposed manuscript describes a preliminary research that must be deeply extended in order to be then considered for publication. The following advices may be helpful for the next submission.
MAJOR CONCERNS:
- No patients’ selection criteria are specified.
- It is not clear whether there are any statically significant differences between the clinical characteristics of the patients enrolled in the two cohorts.
- The third cohort of patients (Bev) does not include any clinical information, so it is impossible to compare with the other two.
- Limited number of selected patients.
Minor issues:
- In case is not possible to include clinical information of the third cohort, I suggest to do not show the data in the graph, but just mention it in the discussion.
- I warmly suggest including another table that compares the clinical features of the patients within the 3 groups (in which data are reported as median plus interquartile range or number and %, according to the parameter) including appropriate statistics. The table should comprise all patient characteristics, such as co-morbidities and all the other parameters useful to clearly address the contribution of vaccination in raising any adverse effects.
Author Response
Point by point response
We are very grateful to the reviewers for their insightful comments and suggestions, which have undoubtedly helped us to improve our manuscript immensely. As indicated in the responses below, we have taken all their comments and suggestions into account when generating the revised version of the manuscript. Responses to the reviewers’ comments appear after the arrows in blue text.
Reviewer 4:
In the manuscript entitled “Adverse events caused by the combined use of bevacizumab and vascular endothelial growth factor receptor-targeted vaccination for the patients with malignant glioma”, the authors investigate adverse event profile in patients treated with Bev after the vaccination in comparison to those treated with Bev collected from the published data, or the vaccination alone. They did not find any significant differences in the adverse event rates between the three groups. Given the clinical relevance of the topic, the proposed manuscript describes a preliminary research that must be deeply extended in order to be then considered for publication. The following advices may be helpful for the next submission.
→
Thank you very much for your comments.
MAJOR CONCERNS:
- No patients’ selection criteria are specified.
→
Thank you for your comment. This is the retrospective study using our three previous clinical trials. In each trial, the inclusion and exclusion criteria were noted in the published paper [Tamura R. BMC Cancer. 2020; Kikuchi R. J Clin Med. 2019; Shibao S. Oncotarget. 2018]. For this retrospective study, the patients treated by Bev after the VEGFRs peptide were included from our three previous clinical trials. Furthermore, patients, who received only VEGFR vaccination (without Bev), were also included from our three previous clinical trials. We have described the following criteria in the section of Materials and Methods.
The inclusion and exclusion criteria were shown in the previous reports [Tamura R. BMC Cancer. 2020; Kikuchi R. J Clin Med. 2019; Shibao S. Oncotarget. 2018].
For this retrospective pilot case-control study, the patients treated by Bev after the VEGFRs peptide vaccination were selected from our three previous clinical trials (May 2012–March 2018). Also, we collected data from patients who received only VEGFR vaccination (without Bev).
- It is not clear whether there are any statically significant differences between the clinical characteristics of the patients enrolled in the two cohorts.
→
As the reviewer indicated, we have added the detailed results in the revised manuscript.
No significant differences were identified in the adverse event rate among the three groups (≥ Grade 2, Vac vs. Vac+Bev, P=0.50; Bev vs. Vac+Bev, P=0.69; Vac vs. Bev, P=0.66: ≥Grade 3, Vac vs. Vac+Bev, P=0.27; Bev vs. Vac+Bev, P=0.73; Vac vs. Bev, P=0.18)
Also, we have added the new Table 1 in the revised manuscript. Statistical analysis was performed to evaluate patient’s clinical characteristics between Vac and Vac+Bev groups.
- The third cohort of patients (Bev) does not include any clinical information, so it is impossible to compare with the other two.
→
No studies assessing the treatment effect for malignant gliomas using both VEGF-A and VEGFR-targeted agents have been reported previously. Therefore, the adverse event profile of the Vac and Vac+Bev groups were compared with the previously reported adverse event profile of the 55 patients with malignant glioma treated by Bev alone (Bev group).
There is a general consensus among experts about Bev-related adverse events. Because our cited paper is the representative one published in Neurology, it has frequently been cited in other papers. However, as the reviewer indicated, the detailed clinical information of Bev group could not be obtained from the paper. We have described the limitation in the section of Discussion.
- Limited number of selected patients.
→
As the reviewer indicated, a limitation of the present study was the paucity of the number of patients treated by Bev after the VEGFRs peptide vaccination. Small sample size decreases statistical power. We have added the limitation in the section of Discussion.
Minor issues:
- In case is not possible to include clinical information of the third cohort, I suggest to do not show the data in the graph, but just mention it in the discussion.
→
According to the reviewer comments. We removed third cohort from the Figure 1.
- I warmly suggest including another table that compares the clinical features of the patients within the 3 groups (in which data are reported as median plus interquartile range or number and %, according to the parameter) including appropriate statistics. The table should comprise all patient characteristics, such as co-morbidities and all the other parameters useful to clearly address the contribution of vaccination in raising any adverse effects.
→
Thank you for your comments. We have added the past histories in Table 2 and 3. None of the patients had a potentially relevant past medical history, such as vasculopathy, in the Vac and Vac+Bev groups.
In addition, we have added the new Table 1 in the revised manuscript. Statistical analysis was performed to evaluate patient’s characteristics between Vac and Vac+Bev groups, because no detailed patient’s characteristics were noted in the previous report of Bev group.
Reviewer 5 Report
The authors have done a very good job with revising the manuscript and I am happy to accept in its current form.
Author Response
Point by point response
We are very grateful to the reviewers for their insightful comments and suggestions, which have undoubtedly helped us to improve our manuscript immensely. As indicated in the responses below, we have taken all their comments and suggestions into account when generating the revised version of the manuscript. Responses to the reviewers’ comments appear after the arrows in blue text.
Reviewer 5:
The authors have done a very good job with revising the manuscript and I am happy to accept in its current form.
→
Thank you very much for your comments.
This manuscript is a resubmission of an earlier submission. The following is a list of the peer review reports and author responses from that submission.
Round 1
Reviewer 1 Report
This is a relatively straightforward and potentially useful short communication. I have the following major points on content and minor language edits for the authors:
- The authors seem to use the terms 'toxicity' and 'adverse events' interchangeably. Please be mindful that drug toxicity describes adverse effects of a drug that occur because the dose or plasma concentration has risen above the therapeutic range, either unintentionally or intentionally (drug overdose). Adverse effects are also not the same as side effects. Adverse events are unintended pharmacologic effects that occur when a medication is administered correctly while a side effect is a secondary unwanted effect that occurs due to drug therapy. It is a common misconception that adverse events and side effects are the same thing. The authors need to be more precise and accurate in the use of terminology throughout the entire manuscript.
- A related point, the title of this report should be modified accordingly. Also, it should be specified in the title that the study population is patients with gliomas.
- The study design allows for limited inferences. A retrospective case-control study design would be more useful in this case.
- "Actually, the simultaneous blockade of both" - Please omit 'actually'.
- In addition to ref 11 and 12, suggest the authors also reference the AVOREN trial (doi:10.1016/S0140-6736(07)61904-7), in which serious adverse events occurred in 29% of patients who received bevacizumab versus 16% of those who did not. Similarly, AEs requiring treatment discontinuation were more frequent in bevacizumab-treated patients versus placebo patients (28% versus 12%).
- Some theoretical background for the efficacy of Bev may be helpful for readers. For example, Bev is thought to be helpful based on the demonstration of high VEGF expression in glioblastoma (GBM) cells as well as in vivo findings (doi: 10.1177/2042098611430109).
- Suggest including some information about significant past medical history (especially vasculopathy, if any) of the study participants in Table 1.
- Please rephrase "It is due to the rarity of Bev administration for the patients following vaccination."
- I would suggest the authors temper the conclusion that this reported showed "the feasibility of the combined use of Bev and VEGFRs peptide vaccination." Such a conclusion must be supplemented by more rigorous evidence. It remains unclear if the findings translate into an overall survival benefit (which may or may not be statistically significant and clinically meaningful).
- The authors should make all data underlying the findings described in their manuscript fully available without restriction at the time of publication. When specific legal or ethical requirements prohibit public sharing of a dataset, authors must indicate how researchers may obtain access to the data.
Reviewer 2 Report
In this manuscript, Tamura et al. have conducted a retrospective study to evaluate the toxicity associated with Bevacizumab (Bev) from their previous VEGF-R peptide vaccine clinical trials conducted in high-grade glioma patients. They show adverse side effect data from patients receiving both VEGF-R vaccine + Bev to point out that including Bev along with VEGF-R vaccine is not associated with any significant toxicity. Though the study shows promise for using Bev for treating patients, as opposed to the previous trials with other solid tumors, there are some issues (listed below) that the authors need to address to strengthen the manuscript.
- Abstract: After reading the first few lines of the abstract or lines 26-28 that explains the adverse side effects from a previously conducted trial, it is not clear what cancer is this manuscript focusing on, though there is an implication that it might be a brain tumor-related clinical trial. Please mention clearly the type of tumor being studied. For example, in lines 22-23, the authors can state that combined use of bevacizumab and VEGF-R peptide vaccine may enhance anti-tumor efficacy in high-grade gliomas.
- Introduction: Please provide more background about high-grade gliomas, current therapy options and why ant-angiogenic therapy is being pursued?
- Introduction: Lines 36-38. Need to elaborate on why targeting angiogenesis is an option for GBM therapy. These two lines are not convincing as a rationale for VEGF pathway-targeting therapy
- Lines 39-43: The authors provide a reference (Ref #3) for their statement about VEGF-targeting inhibitors being used for treating GBM patients and the patients benefits greatly from this therapy. The referenced paper mentions that these inhibitors are being tested in clinical trials and does not state that these inhibitors are used for treating GBMs. Can the authors provide appropriate references for their statement? If not, can the authors rephrase their statement to explain about the real state of affairs with these inhibitors.
- Lines 45-48: Can the authors further explain briefly about the oncoantigens chosen for their previous clinical study?
- Do VEGF inhibitors work well as monotherapies? What is the current state of knowledge regarding VEGF-targeting monotherapies in GBM and other cancers? Please include these details in the introduction.
- Line 52: Please change “Memorized CTL” to “Memory CTL”
- Lines 61-63: Again, please mention that the focus of this study is high-grade gliomas to improve clarity for the readers.
- Materials and methods, line 76: The phrase “Course of vaccine” is not very clear. Does this mean the entire duration that the vaccine/drug was administered? Please provide a better terminology that is widely used for such studies.
- Results: For evaluation of toxicity associated with Bev, the authors have analyzed data only for those patients that received VEGF-R peptide vaccine and Bev. Though it is understandable that they do not see serious issues with toxicity, the authors should also show data from patients that received only VEGF-R vaccine (No Bev). In addition to ruling out issues with toxicity, this comparison between “+Bev” and “No Bev” patients will also provide information if administering Bev is beneficial in any way.
- Results, lines 90-91: “There were no association between type of vaccine, CTL induction, course of vaccine, timing of Bev administration, and complications.” How is this association studied? Please include the relevant data instead of just stating there was no association.
- Was there any indication of improved efficacy from combined VEGF-R peptide vaccine + Bev administration from this study?
- Figure 1: Please elaborate on what the figures mean. What is SAE? Please provide more information about these graphs in the legend.
- How can the authors say that Bev did not result in any statistically significant toxicity when comparing groups of patients that all received Bev – figure 1? Comparison should be between patients that received Bev vs patients that did not receive Bev.
Reviewer 3 Report
In this MS Tamura et al study the toxicity caused by the combined use of bevacizumab and vascular endothelial growth factor receptor targeted vaccination. They found no significant differences in the incidence of Bevacizumab (Bev)-related toxicities between patients with high-grade gliomas treated with and without VEGFRs vaccinations. Although the topic is interesting, the results lack enough consistency to support that there is not additional toxicity by VEGFRs vaccinations. Authors compare their current patients treated with Bev and VEGFRs vaccinations with a previous cohort treated only with Bev. However, they do not explain if both cohorts are comparable. In addition, this second cohort contains 13 patients and, hence, it is difficult to compare with the previous cohort of 55 patients. Moreover, the effect of the treatment, besides a table describing possible toxicities, is not described. Finally, authors describe that there are patients generating CTLs and others do not; however, it is unclear how they measure it. In my opinion this MS is too preliminary to warrant publication in vaccines.